# Real Time Estimation of the Pose of a Lower Limb Prosthesis from a Single Shank Mounted IMU

**DOI:** 10.3390/s19132865

**Published:** 2019-06-27

**Authors:** Clément Duraffourg, Xavier Bonnet, Boris Dauriac, Hélène Pillet

**Affiliations:** 1Arts et Métiers ParisTech, Institut de Biomécanique Humaine George Charpak, 151 Boulevard de l’Hôpital, 75013 Paris, France; 2Proteor^®^, 6 rue de la redoute, 21850 St Apollinaire, France

**Keywords:** lower limb prosthesis, inertial measurement unit, real time, attitude estimation, trajectory reconstruction, strapdown integration

## Abstract

The command of a microprocessor-controlled lower limb prosthesis classically relies on the gait mode recognition. Real time computation of the pose of the prosthesis (i.e., attitude and trajectory) is useful for the correct identification of these modes. In this paper, we present and evaluate an algorithm for the computation of the pose of a lower limb prosthesis, under the constraints of real time applications and limited computing resources. This algorithm uses a nonlinear complementary filter with a variable gain to estimate the attitude of the shank. The trajectory is then computed from the double integration of the accelerometer data corrected from the kinematics of a model of inverted pendulum rolling on a curved arc foot. The results of the proposed algorithm are evaluated against the optoelectronic measurements of walking trials of three people with transfemoral amputation. The root mean square error (RMSE) of the estimated attitude is around 3°, close to the Kalman-based algorithm results reported in similar conditions. The real time correction of the integration of the inertial measurement unit (IMU) acceleration decreases the trajectory error by a factor of 2.5 compared to its direct integration which will result in an improvement of the gait mode recognition.

## 1. Introduction

Over the past decade, prosthetic devices controlled by microprocessor have improved the quality of life of people with lower limb amputation [1]. These devices use different sensors to adapt their behavior to varying terrain. Among these sensors, inertial measurement units (IMUs) can be used to estimate the pose (position and orientation) of a prosthetic segment in order to differentiate between walking situations [2].

To do so, accelerometer and gyroscope data have to be fused. Based on the human segment orientation, fusion algorithms can be classified into three major categories. Kalman-based algorithms are considered as the gold standard [3], but their robustness and real time implementation can be challenging [4]. As an alternative, some methods have been developed to correct the drift of gyroscope integration but are specific to the segment where the IMU is placed, implying strong hypotheses on the segment motion [5,6]. Lastly, a family of methods based on the complementary filter [4,7] have been proposed to quantify the orientation with an accuracy equivalent to the Kalman-based algorithms and an easier real time implementation. This kind of approach has already been evaluated to quantify the orientation of the lower trunk during gait [8].

In the prosthetic field, IMU placement are preferred at the shank [9], which is submitted to high accelerations and impacts during walking. For micro aerial vehicles, nonlinear complementary filters with variable gain have been proposed to limit the repercussion of high accelerations on the orientation estimation [10]. To date, these filters have not been evaluated when tracking the attitude (i.e., roll and pitch) of the lower limb segments during gait.

In addition, from the orientation of the IMU, gravity-free acceleration is integrated to assess the trajectory of the IMU. However, double integration of the acceleration results in exponential drift [11]; hence correction methods are mandatory. When the IMU is placed on the foot, the assumption of zero-velocity during a part of the stance phase is frequently used [12,13,14]. With an IMU placed at the shank, additional kinematic hypothesis is necessary to assess the velocity at a specific time event and correct the integration [2,15]. To our knowledge, no previous study reported the accuracy of the trajectory of shank points obtained with these techniques compared to the direct optoelectronic motion capture.

The aims of this study are: (1) To evaluate the estimation of a prosthetic shank attitude from a nonlinear complementary filter with a variable gain; and (2) to assess and evaluate the trajectory of a prosthesis point using a robust kinematic model of the lower limb during stance. The proposed method was designed to allow real time application with low computational resources. The pose estimated from a single IMU placed on the prosthetic shank of three people with amputation during gait were processed. The pose obtained from optoelectronic measurement was used as a reference for comparison purpose.

## 2. Materials and Methods

### 2.1. Attitude Estimation

The definition of the IMU frame used in this study to determine the attitude of the shank are presented in Figure 1. Euler angles were computed from the angular position matrix of the IMU frame relative to a global earth-fixed frame. A sequence z y’ x’’ (mobile axis) was chosen, which gave the nautical angles (i.e., yaw, pitch, and roll) and prevented any effect of the yaw estimation error on the other two angles.

The attitude (pitch and roll) of the prosthesis was obtained using a nonlinear complementary filter with variable gain. This filter, adapted from Valenti et al., uses the Equation (1) to fuse angular estimations from the accelerometer and from the gyroscope integration. This filter was chosen because it can easily be implemented without a magnetometer, and with low computational power [10].
(1)$$q_{t}=\left(q_{t-1}+{\dot{q}}_{\omega }\Delta t\right)⊗\left(\left(1-\alpha \left(e\right)\right)q_{I}+\alpha \left(e\right)\Delta q_{acc}\right)$$

In this equation, all the terms are expressed in the global earth-fixed frame. $q_{t}$ is the quaternion representative of the attitude estimation at time step t, ${\dot{q}}_{\omega }$ is the quaternion representing the angular velocity from the gyroscope and Δt is the elapsed time since the last integration. The first part of the equation corresponds to the integration of the gyroscope between two time steps using the trapezoidal rule. In the second part of this equation, $\Delta q_{acc}$ represents the orientation of the gravity relatively to its estimation using the data from the accelerometer. $q_{I}$ is the identity quaternion. This second part corresponds to a low pass filtering of $\Delta q_{acc}$ which time constant depends on a gain α(e).

α(e) varies according to the error e computed as the normalized difference between the norms of the acceleration and the gravity. For errors lower than a first threshold $th_{1}$, α(e) equals $\alpha_{cst}$, and for errors higher than a second threshold $th_{2}$, the gain is set to zero. In between the two thresholds the gain decreases linearly as a function of the error.

During the swing phase of gait, the acceleration can be of magnitude close to the one of the gravity vector but with very different orientation, leading to the computation of a null error e. To account for this particularity, a penalization term (swing) was added in order to discard accelerometer measurements during the swing phase (Equation (2)).
(2)$$e=\frac{\left|||\left[\vec{a}\right]||-g\right|}{g}+swing$$

swing equals one during the swing phase and zero otherwise.

Finally, without magnetometer, it is not possible to correct the drift of the yaw angle. This angle was simply reset at the beginning of each gait cycle. 

The numerical values used in this study are given in Table 1. 

### 2.2. Trajectory Estimation

In the present study, we chose to compute the trajectory of the center of the knee (K Figure 2). It should be noted however that the trajectory of any point of the shank could be obtained in a similar way. The proposed method combines the use of a kinematic model during unipodal stance and double integration of the acceleration of the IMU during swing. The model used is represented in Figure 2. 

During the unipodal stance, the velocity of K was obtained by the means of a model of an inverted pendulum rolling on a curved arc foot (Figure 2). The velocity of the center of the arc (${\left[\vec{V_{C}}\right]}_{R_{0}}$) (C Figure 2) was computed with a rolling without sliding hypothesis using reference data from Hansen et al. to define the round shape dimensions [16]. The velocity of K (${\left[\vec{V_{Kstance}}\right]}_{R_{0}}$) was finally obtained in the global earth-fixed frame using the attitude estimation obtained (Equation (3)).
(3)$${\left[\vec{V_{Kstance}}\right]}_{R_{0}}={\left[\vec{V_{C}}\right]}_{R_{0}}+{\left[\vec{KC}\right]}_{R_{0}}\wedge {\left[\vec{\Omega }\right]}_{R_{0}}$$

During the swing phase, the accelerometer data were projected in the global earth-fixed frame to remove the gravity acceleration. The acceleration of the IMU (${\left[\vec{A_{I}}\right]}_{R_{0}}$) (I Figure 2) was then used to compute the acceleration of K (${\left[\vec{A_{K}}\right]}_{R_{0}}$) using Equation (4).
(4)$${\left[\vec{A_{K}}\right]}_{R_{0}}={\left[\vec{A_{I}}\right]}_{R_{0}}+{\left[\vec{KI}\right]}_{R_{0}}\wedge \frac{d{\left[\vec{\Omega }\right]}_{R_{0}}}{dt}+\left({\left[\vec{\Omega }\right]}_{R_{0}}\wedge {\left[\vec{KI}\right]}_{R_{0}}\right)\wedge {\left[\vec{\Omega }\right]}_{R_{0}}$$

${\left[\vec{A_{K}}\right]}_{R_{0}}$ is then integrated to obtain the velocity of K during the swing phase ${\left[\vec{V_{Kswing}}\right]}_{R_{0}}$.

Figure 3 illustrates the method on an example of evolution of the antero-posterior component of ${\left[\vec{V_{Kswing}}\right]}_{R_{0}}$ and ${\left[\vec{V_{Kstance}}\right]}_{R_{0}}$

The difference at the beginning of unipodal stance (US) between both the estimations of ${\left[\vec{V_{K}}\right]}_{R_{0}}$ was computed and referred hereafter as ${\left[\vec{V_{Kdiff@US}}\right]}_{R_{0}}$ (Figure 3). This error was used for the correction of the integration assuming a constant bias on the swing acceleration $\frac{{\left[\vec{V_{Kdiff@US}}\right]}_{R_{0}}}{T}$ where T is the duration of integration. 

A first method for the correction removes this bias from the obtained swing velocity a posteriori (i.e., correction of the previous swing phase) using Equation (5). The corrected velocity is referred to as ${\left[\vec{V_{Kpost}}\right]}_{R_{0}}$ (Figure 3).
(5)$${\left[\vec{V_{Kpost}\left(t\right)}\right]}_{R_{0}}=\overset{t}{\underset{t-T}{\int }}{\left[\vec{A_{K}}\right]}_{R_{0}}dt-t\times \frac{{\left[\vec{V_{Kdiff@US}}\right]}_{R_{0}}}{T}$$

A second method updated a correction term at each *i*th cycle (${\left[c_{i}\right]}_{R_{0}}$) according to Equation (6). 

(6)$${\left[c_{i}\right]}_{R_{0}}={\left[c_{i-1}\right]}_{R_{0}}+K\frac{{\left[\vec{V_{Kdiff@US}}\right]}_{R_{0}}}{T}$$

${\left[c_{i}\right]}_{R_{0}}$ can be used to correct a priori (i.e., in real time) the integration at each time step (t) using Equation (7). The term K is a constant meant to avoid divergence of the correction term when $\frac{{\left[\vec{V_{Kdiff@HS}}\right]}_{R_{0}}}{T}$ is noisy. In this study, we set K=0.8 to impose a quick convergence. For long-term acquisitions, this constant should be lower. 

The obtained velocity is referred to as ${\left[\vec{V_{Kprior}}\right]}_{R_{0}}$. The Figure 4 illustrates the effect of this correction. 

(7)$${\left[\vec{V_{Kprior}\left(t\right)}\right]}_{R_{0}}={\left[\vec{V_{Kprior}\left(t-1\right)}\right]}_{R_{0}}+\overset{t}{\underset{t-1}{\int }}\left({\left[\vec{A_{K}}\right]}_{R_{0}}-{\left[c_{i}\right]}_{R_{0}}\right)dt$$

The last step to obtain the trajectory is the direct integration of the corrected velocity by applying a zero reset to the position at US. In this study, all the integrations are performed using the trapezoidal rule.

### 2.3. Experiments

These algorithms are applied on the data of three people with transfemoral amputation following a protocol approved by the Ethics Committee (Comité de Protection des Personnes CPP NX060336). A total of 4, 20 and 12 gait cycles were extracted for participants 1, 2, and 3 respectively. Hence a total of 36 gait cycles were considered. The participants were asked to walk at their self-selected speed on the ground level. Anthropometric data of the three participants are presented in the Table 2.

These participants were equipped with a custom datalogger strapped onto their prosthesis. This datalogger used a microcontroller (Arduino nano, Arduino^®^) to transmit the data from a low cost IMU (MPU6050, InvenSense inc.^®^, $0.8) to a laptop using a Bluetooth connection. The Bluetooth connection and data collection were managed on the laptop using Matlab software (Matlab R2016b, MathWorks^®^). The data were sampled at 100 Hz.

The reference measurement of the shank pose was obtained from an optoelectronic motion capture (MOCAP) system (Vicon, Oxford, UK). The orientation of the shank was derived from the markers attached to the shank, and the trajectory of the knee was assumed to be the mean trajectory of lateral and medial condyles.

### 2.4. Data Analysis 

For the orientation, the root mean square errors (RMSE) of the estimation of the attitude (roll and pitch) provided by the method described in this article relative to MOCAP was computed for each gait cycle. Similarly, we computed the RMSE of the estimation of the trajectory of the knee joint center. Average, minimum, and maximum were then calculated across all gait cycles for each participant.

In order to compare our results with the literature, we computed the difference between the estimations of the stride length from the knee trajectory, obtained with the IMU and with MOCAP, at each gait cycle. This error was then normalized using the MOCAP estimation and averaged across all gait cycles. Extremum values were also extracted for each participant.

## 3. Results

Figure 5 shows the average attitude across all gait cycles, and the envelope containing all curves for each participant according to the gait cycle and for both methods (IMU-based and MOCAP). The errors on the attitude estimation are reported in Table 3. The RMSE on the trajectory are given in Table 4, and the stride length errors are given in Table 5. For all errors presented results includes mean across all cycles of the considered participant, as well as minimum and maximum.

## 4. Discussion

### 4.1. Algorithm Evaluation

The aims of the study are: (1) To evaluate the estimation of prosthetic shank attitude from a nonlinear complementary filter with a variable gain; (2) to assess and evaluate the trajectory of the prosthetic knee using a robust kinematic model of the lower limb during stance. The proposed method was designed to allow real time application with low computational resources.

In the literature, Kalman-based algorithms are often taken as a reference for attitude estimation using IMU with RMSE on the orientation of the trunk during gait reported to be as small as 1° [17]. Yet this precision decreases for segments submitted to rapid movements with large variations of centripetal acceleration [18] and can lead to RMSE around 3° for the shank [19,20]. In addition, in the context of estimation of the attitude of a prosthetic device, Kalman-based algorithms are computationally expensive which limits their real time implementation. In contrast, in this study, a nonlinear complementary filter and a variable gain strategy adapted from Valenti et al. was used to obtain the attitude of the prosthetic shank with a low computational cost [10]. Overall, the RMSEs were 2.8° for the pitch angle and 2.4° for the roll angle for gait at an average speed of 4 km/h. Thus, the attitude estimation is close to the one obtained from the Kalman filter but requires much less computational power, making onboard real-time implementation easier. Moreover, this algorithm overcomes the intrinsic limitations of the Kalman filter, such as the need for a high sampling rate, and linearization issues [4]. 

For trajectory estimation, few studies assessed the full trajectory of the foot and evaluated this trajectory qualitatively. Quantitative evaluation was only performed on parameters extracted from this trajectory such as the mean error on the stride length. Most of these studies used an IMU placed on the foot and assumption of zero velocity update (ZVU) to correct the integration a posteriori. Mariani et al. reported an error on the stride length estimation of 1.3 ± 6.5% (mean ± standard deviation) [14]. With an IMU placed on the tibia, the ZVU hypothesis is no longer relevant [21]. Yang and Li used direct integration of the gravity free acceleration and obtained an error between 13.5% and 21.5%. They also adjusted their results using a linear regression model based on one point estimation of the velocity and the reference step length. This strategy reduced this error between 2.4% and 5.4% [22].

In the present study, we proposed a model of the movement of the prosthetic shank represented by an inverted pendulum rolling on a curved arc foot during the whole unipodal stance. This kinematic model gave an estimation of the linear velocity of the tibia during stance, which can be used to correct the integration. When the correction was performed a posteriori, the stride length was estimated with an error of −5.1 (−12.5/1.0) % (mean (min/max)) of the stride length. The non-zero mean indicate a slight underestimation of the stride length. Compared to the literature, the a posteriori correction gives results similar to ZVU strategies and close to the adjusted results of Yang and Li, which suggests that the model on which the trajectory estimation is based is a good alternative to the ZVU when the IMU is placed on the tibia.

It should be noted however that for one subject, the stride length error was higher. This can be explained by the fact that this participant had a very particular gait pattern that might not be represented by the inverted pendulum rolling on a curved arc foot model especially when using an average round shape of the curved arc foot. Personalization of this model would probably result in a more accurate trajectory estimation. For a prosthetic foot this round shape mainly depends on the prosthetic foot design and its alignment [16], personalization could therefore be made using the characteristics of the prosthetic foot and a measurement of the foot alignment.

In the context of prosthetic control, we need to obtain an a priori estimation of the kinematics. In this study, a correction at each frame of the integration is proposed which is an alternative of direct integration used in the literature to obtain trajectory during swing phase in real time [2,23]. The results obtained with this algorithm allow a decrease in the average error and in the dispersion by a 2.5 factor compared to a direct integration (Table 5). Looking at the RMSE on the full trajectory, it appears that the estimation is improved in both axes by a fair amount (Table 4).

### 4.2. Limitations 

The computation of ${\left[\vec{V_{Kdiff@US}}\right]}_{R_{0}}$ might not be optimal. In the presented algorithm, ${\left[\vec{V_{Kdiff@US}}\right]}_{R_{0}}$ is computed at the start of the unipodal stance, after the heel strike, thus the dynamic effects probably affects this estimation of the drift due to the first integration of the gravity free acceleration. Bergamini et al. reported that the drift due to numerical integration might depend on the amplitude of the integrated data [8]. This is also supported by a better estimation of stride length using a nonlinear correction developed by Mariani et al. compared to a linear one [14]. A possible improvement is to take advantage of the model to calculate an error using the whole unipodal stance. The correction could moreover be modulated according to the amplitude of the integrated data.

The results presented herein are obtained with data from a low cost IMU (MPU 6050, InvenSence, 0.8 $). When compared to other commercially available sensors (MT1i 1-series, Xsens), this sensor shows limitations. Specifically, the noise density of the accelerometer and the nonlinearity of the gyroscope are twice that of Xsens sensor. There are also differences in terms of zero rate and zero *g* output [24,25]. For attitude estimation the noise density of the gyroscope is similar for both sensors and the zero rate output can be removed easily. For trajectory computation however, the higher noise density on accelerometer data favors the accumulation of errors during the successive integrations. 

The evaluation of the algorithms was based on the data obtained from only three people with transfemoral amputation. This small number of participation is due to the difficulties to recruit people with transfemoral amputation presenting good walking abilities (no walking aid). Due to tiredness (from other acquisitions) of the 1st participant, acquisitions were stopped, hence he only completed four gait cycles. The conclusions presented herein are tainted with individual bias and should be taken as preliminary results.

It should be noted, however, that very few studies have tested their algorithms on amputated people for whom the impacts during gait are not filtered by soft tissues. Moreover these studies often recruit only a limited number of participants with lower limb amputation (e.g., [2,23,26,27,28,29]).

For people with above knee amputation, the ferromagnetic materials and the presence of motors in microprocessor prosthetic leg makes magnetometer unusable [30]. Hence in this study the yaw angle could not be corrected. However, attention was paid to the choice of the filter and the axis sequence for the calculation of the angles. For trajectory reconstruction it has probably only a limited effect on the trajectory in the sagittal plane during straight line walking but it might be of importance for out of plane ambulation. More work would be necessary to obtain the full 3D trajectory, including correction of the yaw angle and computation of a reference velocity in the frontal plane. However the trajectory in the sagittal plane is usually sufficient for prosthetic control [2,23].

The presented algorithm is primarily designed for prostheses control (i.e., real time activity recognition), but it can also be useful for orthosis, exoskeleton, or activity monitoring device. This algorithm could also be adapted for an IMU placed more proximally with a modification of the model describing the unipodal stance phase kinematics. Moreover its application range is not limited to activity recognition, it could also be used for activity monitoring of specific pathologies as suggested in [31].

## 5. Conclusions

In this study an algorithm was developed for real time pose estimation, with consideration for computation power limitation in an embedded system. The results presented are obtained owing to walking trials of three people with transfemoral amputation using one low cost IMU on the shank. For attitude estimation, a nonlinear complementary filter with a variable gain strategy was used and showed results close to Kalman-based algorithms. This study suggests that this type of filter is suitable for prosthetic lower limb attitude estimation.

The velocity computation is based on the integration of the accelerometer data corrected owing to a model of inverted pendulum rolling on the curved arc foot. The trajectory is obtained owing to direct integration of this velocity. Two methods of correction of the integrated velocity are evaluated. A posteriori correction results are close to the literature. The a priori method decreased the error by 2.5 regarding the direct integration. This algorithms should allow a better real time estimation of the trajectory having the potential to permit a faster gait mode detection [26]. 

Future work will focus on the improvement of the trajectory drift estimation by taking more advantage of the motion model during the unipodal stance as discussed previously. Implementation on prosthesis and test during non-level ambulation for gait mode detection is also planned.

## Figures and Tables

**Figure 1 sensors-19-02865-f001:**
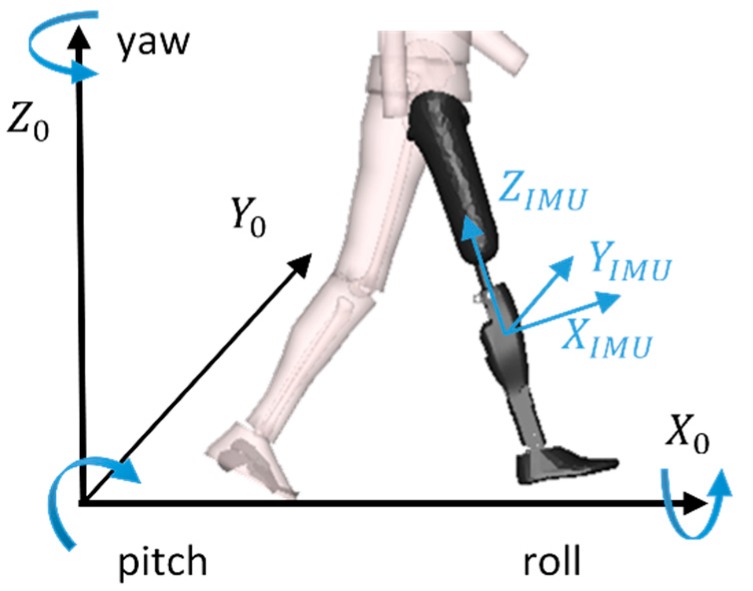
Definition of the inertial measurement unit (IMU) and global earth-fixed frames and representation of the three angles defining the orientation of the IMU relative to the global frame.

**Figure 2 sensors-19-02865-f002:**
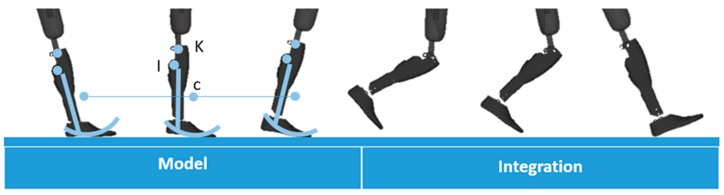
Computation strategy and illustration of the model used. C is the center of the curved arc foot model, K is the center of the knee, I is the center of the IMU frame.

**Figure 3 sensors-19-02865-f003:**
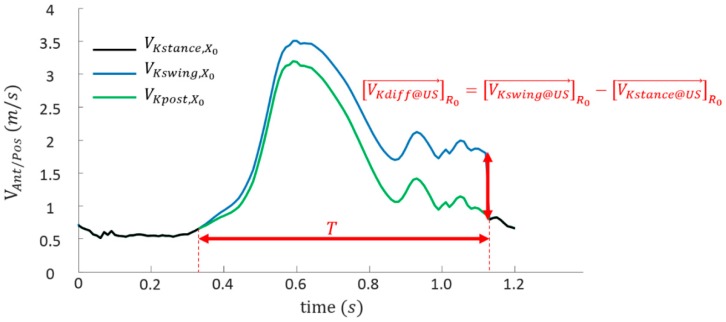
Illustration of the computation of the error ${\left[\vec{V_{Kdiff@US}}\right]}_{R_{0}}$ and the effect of a posteriori correction on an exemple of antero-posterior component of the knee velocity.

**Figure 4 sensors-19-02865-f004:**
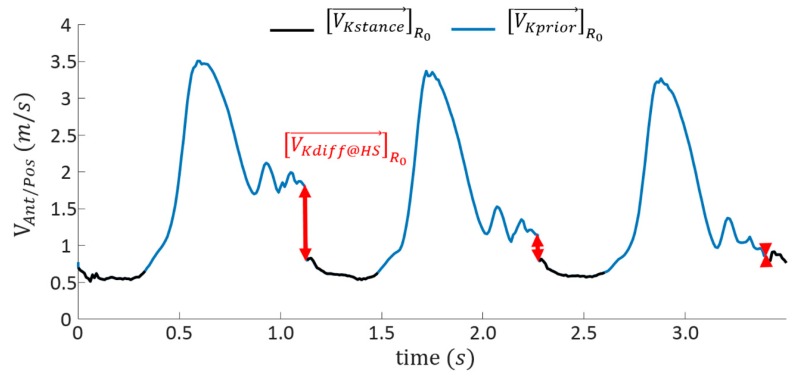
Illustration of a priori correction on an example of anteroposterior component of the knee velocity.

**Figure 5 sensors-19-02865-f005:**
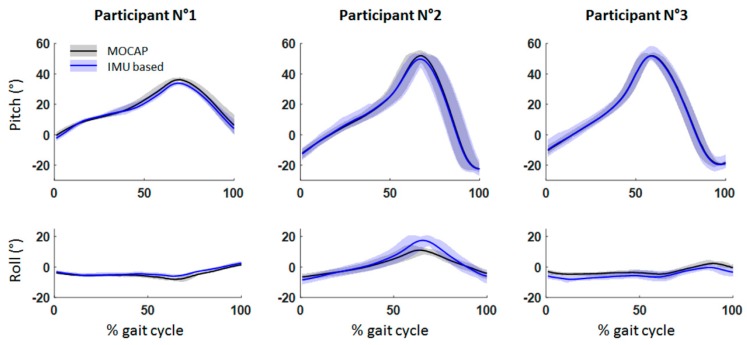
Attitudes obtained from the motion capture (MOCAP) system (black) and the IMU (blue). The mean curve (solid line) and the envelope containing all curves are shown for each participant.

**Table 1 sensors-19-02865-t001:** Numerical constants used in the study.

Variable	Numerical Value
Δt (s)	0.01
$\alpha_{cst}$	0.02
$th_{1}$	0.1
$th_{2}$	0.15

**Table 2 sensors-19-02865-t002:** Anthropometric data of the participants of the present study.

Participant	N°1	N°2	N°3
Height (m)	1.75	1.72	1.75
Weight (kg)	57	98	95
Gender	M	M	M

**Table 3 sensors-19-02865-t003:** Attitude estimation for each participant (mean(min/max)). The roll and pitch angles are in the frontal and sagittal planes respectively.

Participant	Gait Speed (km/h)	Number of Gait Cycles	RMSE Roll (°) Mean (min/max)	RMSE Pitch (°) Mean (min/max)
N°1	2.64	4	1.32 (0.82/1.82)	1.98 (1.49/2.49)
N°2	4.74	20	3.19 (2.57/5.63)	2.47 (1.30/5.12)
N°3	4.79	12	2.54 (1.53/3.87)	2.51 (0.97/5.32)
Overall	4.06	36	2.77 (0.82/5.63)	2.43 (0.97/5.32)

**Table 4 sensors-19-02865-t004:** Root mean square error (RMSE) using direct integration, a priori correction and a posteriori correction from IMU data compared to MOCAP data.

Trajectory Error	Participant	A Posteriori Correction Mean (min/max) (cm)	A Priori Correction Mean (min/max) (cm)	Direct Integration Mean (min/max) (cm)
RMSE along X	N°1	2.2 (0.9/3.3)	3.6 (0.9/7.5)	239.4 (40.4/350.5)
	N°2	3.0 (1.7/7.0)	3.8 (1.0/6.5)	163.1 (3.9/389.4)
	N°3	2.8 (1.1/3.9)	4.0 (1.1/6.6)	171.8 (115.4/808.6)
	overall	2.8 (0.9/7.0)	3.2 (0.9/7.5)	233.9 (3.9/808.6)
RMSE along Z	N°1	1.8 (1.5/2.2)	2.3 (1.8/3.6)	18.5 (2.8/25.2)
	N°2	2.0 (0.7/4.8)	2.8 (1.0/7.0)	39.0 (19.5/75.9)
	N°3	2.1 (1.9/9.8)	2.6 (1.0/4.4)	31.4 (37.6/139.0)
	overall	2.8 (0.7/9.8)	2.8 (1.0/7.0)	55.2 (2.8/139.0)

**Table 5 sensors-19-02865-t005:** Errors on the stride length estimation. Mean stride length error and % RMSE are computed using the estimation of the stride length of each gait cycle.

Stride Length Error	Participant	A Posteriori Correction (% Stride Length)	A Priori Correction (% Stride Length)	Direct Integration (% Stride Length)
Mean(min/max) stride length error	N°1	−8.4 (−12.5/−3.4)	−13.0 (−28.3/−1.3)	2.0 (−4.2/8.7)
	N°2	−5.2 (−11.8/1.0)	−5.8 (−12.5/4.4)	−16.4 (−26.5/−5.4)
	N°3	−3.6 (−6.7/−0.5)	4.3 (−3.3/13.4)	−31.0 (−48.0/−20.4)
	overall	−5.1 (−12.5/1.0)	−3.6 (−28.3/13.4)	−18.6 (−48.0/8.7)

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
