# Peer review of "Real Time Estimation of the Pose of a Lower Limb Prosthesis from a Single Shank Mounted IMU"

_sensors, 2019, doi:10.3390/s19132865_

Round 1

Reviewer 1 Report

Authors developed and validated an algorithm for the computation of the pose of a lower limb prosthesis. Although, the manuscript is generally reader-friendly and well presented, but this reviewer identified some major missing information or problems with the study design, those are pointed here.

1. First, there is a doubt that the validated result would be related to individual characteristic because of small number to participate, just 3 subjects (36 gait cycles) in this study. Thus, the study with 3 subjects (less gait cycles) are not enough for drawing a robust conclusion. Authors should use more subjects and re-do their analysis. In the Table 2, you have mentioned that subjects are patient, this is really unclear as first time used in the manuscript. What type of patient are they? Additionally, it was applied only on the male subjects, but if there is/are female subjects added, results might be change (gait patterns are different Male Vs. Female).

2. Only RMSE result is not sufficient to validate the algorithm.

3. The implications and relevance of the work are currently concealed due to a poor framing of the problem space and very limited discussion of the findings and their relevance.

4. has this IMU device been validated before against lab standards

5. Were the data normally distributed? Was there a check for this?

6. Define all the acronyms, first time that they are used Like, IMU, RMSE .

7. The entire Recommendation section also consists of blanket statements that do not actually provide any useful new information to researchers.

8.  In the figure 3 and 4, what are the units in the X and Y axis?

9.  Of the references used, several rather old (more than 5 years) papers are used to support important statements.

Author Response

Response to Reviewer 1 Comments

We express our sincere thanks to the reviewer for his/her careful reading of the manuscript and his/her very helpful and constructive comments. The manuscript has been considerably improved following these remarks. The responses at each reviewer comment are given hereafter. The modifications are highlighted in the manuscript.

1. First, there is a doubt that the validated result would be related to individual characteristic because of small number to participate, just 3 subjects (36 gait cycles) in this study. Thus, the study with 3 subjects (less gait cycles) are not enough for drawing a robust conclusion. Authors should use more subjects and re-do their analysis. In the Table 2, you have mentioned that subjects are patient, this is really unclear as first time used in the manuscript. What type of patient are they? Additionally, it was applied only on the male subjects, but if there is/are female subjects added, results might be change (gait patterns are different Male Vs. Female).

Thank you for this relevant comment. Even if the robustness of the proof is directly related to the number of participants included in the study, this relatively small number must be put in perspective compared to existing published datasets.

Indeed, such a small population is not uncommon in studies focusing on a first assessment of a prosthesis design on people with amputation. The following references are examples of such studies (TT stand for Transtibial amputation and TF for transfemoral amputation) : [1] 5TT, [2] 6TT, [3] 1TF, [4] 2TF, [5] 1TF, [6] 5TF, [7] 1TF, [8] 3TF. .

1. Mariani, B. Assessment of Foot Signature Using Wearable Sensors for Clinical Gait Analysis and Real-Time Activity Recognition. Phd Thesis, École polythechnique fédérale de Lausanne: Lausanne, Switzerland, 2012.

2. Stolyarov, R.; Burnett, G.; Herr, H. Translational Motion Tracking of Leg Joints for Enhanced Prediction of Walking Tasks. IEEE Trans. Biomed. Eng. 2018, 65, 763–769.

3. Seel, T.; Raisch, J.; Schauer, T. IMU-Based Joint Angle Measurement for Gait Analysis. Sensors 2014.

4. Seel, T.; Graurock, D.; Schauer, T. Realtime assessment of foot orientation by Accelerometers and Gyroscopes. Curr. Dir. Biomed. Eng. 2015.

5. Lawson, B.E.; Varol, H.A.; Goldfarb, M. Standing stability enhancement with an intelligent powered transfemoral prosthesis. IEEE Trans. Biomed. Eng. 2011.

6. Huang, H.; Zhang, F.; Hargrove, L.J.; Dou, Z.; Rogers, D.R.; Englehart, K.B. Continuous locomotion-mode identification for prosthetic legs based on neuromuscular - Mechanical fusion. IEEE Trans. Biomed. Eng. 2011, 58, 2867–2875.

7. Varol, H.A.; Sup, F.; Goldfarb, M. Multiclass real-time intent recognition of a powered lower limb prosthesis. IEEE Trans. Biomed. Eng. 2010.

8. Lawson, B.E.; Mitchell, J.; Truex, D.; Shultz, A.; Ledoux, E.; Goldfarb, M. A robotic leg prosthesis: Design, control, and implementation. IEEE Robot. Autom. Mag. 2014, 21, 70–81.

In our sample, a significant effect of the individual has been observed through one way analysis of variance, showing that the results should be regarded as preliminary results. However, the recruitment of people with amputation is a real issue which limits the possibility of extensive study.

Despite these limitations, this preliminary study remains useful and valuable as it is performed on people with amputation and the signals recorded by the IMU therefore present the characteristics relative to this specific population. In particular, the absence of soft tissues implies direct transmission of mechanical load in the prosthesis (made mainly of carbon fiber and aluminum). During gait the impact are therefore transmitted entirely to the IMU leading to different needs in the filtering process than for signals obtained on non-amputee subjects. This justifies the choice to not include other participants such as non-amputee subjects easier to recruit but that will not lead to realistic signals in the context of prosthetic design.

The limitation part of the article was modified to include these remarks as follows:

The evaluation of the algorithms was based on data obtained from only 3 people with transfemoral amputation. This small number of participants is due to the difficulties to recruits people with transfemoral amputation presenting good walking abilities (no walking aid). Due to tiredness (from other acquisitions) of the 1st participant acquisitions were stopped, hence he only completed 4 gait cycles. The conclusions presented herein are tainted with individual bias and should be taken as preliminary results.

Also the first version of the article may have been a bit too affirmative considering the level of proof, the conclusion of the reviewed version was modified, replacing shows with suggests at line 275.

The word “Patient” in table 2 was a remnant of previous version, it was changed to participant in the reviewed paper. The participants in this study did not present any disability affecting the gait other than amputation (this was checked by a medical doctor).

As concerns the differences between male and female, we agree with the reviewer that it would be interesting to check the effect of gender on these algorithms but we think that they are probably limited compared with lower limb amputation. Moreover the difficulties to recruit volunteers increases a lot when seeking female subject with lower limb amputation due to the low incidence of amputation among females

2. Only RMSE result is not sufficient to validate the algorithm.

Thank you for your remark, we agree that only the RMSE is not enough to validate the algorithm. Please note that wedid not claim to validate the presented algorithms, this study deals primarily with the design of algorithms to obtain orientation and position of the shank. An evaluation of these algorithms on a reduced sample of the target population is provided as preliminary results.

When looking at the literature, studies developing algorithms for the estimation of orientation from IMUs usually evaluate their algorithm only through RMSE, see for instance the following references: 

Valenti, R.G.; Dryanovski, I.; Xiao, J. Keeping a good attitude: A quaternion-based orientation filter for IMUs and MARGs. Sensors (Switzerland) 2015.

Madgwick, S.O.H.; Harrison, A.J.L.; Vaidyanathan, R. Estimation of IMU and MARG orientation using a gradient descent algorithm. In Proceedings of the 2011 IEEE International Conference on Rehabilitation Robotics; IEEE: Zurich, 2011; pp. 1–7.

Tian, Y.; Wei, H.; Tan, J. An adaptive-gain complementary filter for real-time human motion tracking with MARG sensors in free-living environments. IEEE Trans. Neural Syst. Rehabil. Eng. 2013.

For the reconstruction of the trajectory of a point of the device,  the authors is not aware of any study reporting error on the full trajectory, we thus choose to compute the RMSE to be consistent with the way orientation errors were characterized.

In the literature, most of the studies evaluating algorithms for strapdown integration quantified the error on the stride length estimation or on the total distance walked as in the references below. For the sake of comparison with these studies, we also compute an error on the stride length estimation.

Mariani, B.; Hoskovec, C.; Rochat, S.; Büla, C.; Penders, J.; Aminian, K. 3D gait assessment in young and elderly subjects using foot-worn inertial sensors. J. Biomech. 2010, 43, 2999–3006

Yun, X.; Bachmann, E.; Moore, H.; Calusdian, J. Self-Contained Position Tracking of Human Movement Using Small Inertial/Magnetic Sensor Modules. In Proceedings of the 2007 IEEE International Conference on Robotics and Automation; Roma, Italy, 2007; pp. 2526–2533.

Yang, S.; Li, Q. IMU-based ambulatory walking speed estimation in constrained treadmill and overground walking. Comput. Methods Biomech. Biomed. Engin. 2012.

3. The implications and relevance of the work are currently concealed due to a poor framing of the problem space and very limited discussion of the findings and their relevance.

Thank you for this remark. The authors are not sure of what refers to the sentence « a poor framing of the problem space». The authors understood that the field of application of the algorithm was narrowed due to the phrasing of the setting up of the problem and the lack of explicit perspectives.

The authors purposely reduced the problem space in the introduction to focus on the specific challenges related to prosthesis control (real time, low computational power, high accelerations). The reviewer is however right to point that this work could be useful for other application, a paragraph was added at the end of the discussion to broaden the possible field of application of this algorithm as follows :

The presented algorithm is primarily designed for prostheses control (i.e. real time activity recognition), but it could also be useful for orthosis, exoskeleton, or activity monitoring device. This algorithm could also be adapted for an IMU placed more proximally with a modification of the model of the kinematic during the unipodal stance phase. Moreover its application range is not limited to activity recognition, it could also be used for activity monitoring of specific pathologies as suggested in [34].

4. Has this IMU device been validated before against lab standards

Thank you for this interesting remark. In this study we deliberately used a low cost IMU taken off the shelf as the aim of the work was to develop algorithms for prosthetic control that would use that kind of turnkey sensor. We did not assess the accuracy of the raw measurements of this sensor neither did we performed a calibration procedure. This IMU (MPU 6050) is however a well-known component with comprehensive datasheet and that have been used in other studies before (e.g. reference hereafter).

Magnussen, O.; Ottestad, M.; Hovland, G. Experimental validation of a quaternion-based attitude estimation with direct input to a quadcopter control system. 2013 Int. Conf. Unmanned Aircr. Syst. ICUAS 2013 - Conf. Proc. 2013, 480–485.

The sensor accuracy seems to have a limited effect on the attitude estimation, since the errors are close to studies using reference measurement and algorithm. For trajectory estimation the sensor inaccuracy may indeed take a part in the observed errors. Please note that the purpose of this study was to develop and evaluate algorithms, the comparison between algorithms remains relevant, even if the measurements of the sensor are biased.

5. Were the data normally distributed? Was there a check for this?

We fully agree that the use of standard deviation was not adapted for non-normal distribution. In the new version of the paper we, therefore, replace it with the min and max error across all cycles for each participant.

6. Define all the acronyms, first time that they are used Like, IMU, RMSE.

Thank you for pointing that out. IMU has been replaced by Inertial Measurement Unit (IMU) in the abstract. We leaved IMU in the title as it would increase a lot the length of the title considering that this acronym is common enough to be understandable.

7. The entire Recommendation section also consists of blanket statements that do not actually provide any useful new information to researchers.

The authors are sorry, as they don’t understand this comment. To what section does the reviewer refer? Both the Sensors template and the instruction for authors does not refer to a recommendation section, and neither does the manuscript include such section.

8.  In the figure 3 and 4, what are the units in the X and Y axis?

Thank you for this remark. The units in these figures were concealed due to a bad placement of the legends. We fixed this in the reviewed paper by placing axis legends at the left of the vertical axis and under the horizontal axis.

9.  Of the references used, several rather old (more than 5 years) papers are used to support important statements.

Thank you for this comment, references used in this study are indeed rather old. This is mainly due to the sparse literature regarding estimation of the pose of a lower limb segment using IMUs during gait. The authors are unaware of more recent similar studies that could be used to further support this work.

Reviewer 2 Report

This study introduces a method of pose estimation for a lower limb prosthesis using an IMU positioned on the shank.  The results suggest error on par with what has previously been reported in the literature.  My comments are related mostly to the wording.  

My one major concern is that only three participants were used as test subjects, and one of them only had four gait cycles.  I think this needs to be addressed more comprehensively in the limitations section.  What are the implications for having such a small test group and limited sample from the individual with the greatest error?

Also, in Line 209 it is stated that this is a “good estimation” – please define good and indicate the implications of error.  How does error influence the gait of people with a lower limb prosthesis?  How much error can be tolerated?

·       Line 47: “The” should be “the”

·       Line 52: “technics” should be “techniques”

·       Line 55 and 178: Consider “… variable gain, and 2/ to assess …”

·       Line 140: Consider indicating here how many gait cycles for each participant.  At first it is unclear if it is 36 for each or 12 for each (for a total of 36).  But then in the table on the next page it shows a variable number of gait cycles for each.  It would be good to include this information in this line.

·       Line 144: What does “10PG”, “11TG” and “12JG” mean?  Are they subject IDs?  For the purpose of the publication, it would be best to call them participant 1, 2, and 3.  If the numbers/letters stand for something relevant to the paper, those abbreviations should be defined.  Also, in this table they are referred to as “patients” but “participants” in other tables, and “people” at other times in the text.  It would help to be consistent.

·       Line 147 and 229: the $ should go before the number ($0.80 is common).

·       Line 155: “relatively” should be “relative”

·       Line 165: “Stride” should be “stride”

·       Line 165: It seems a little odd to include 95% confidence intervals for something with just four cycles.  Is it possible to somehow show all four cycles for this participant instead?

·       Line 170: In this table, I assume that the RMSE was calculated over the entire cycle (i.e. error determined for all points in the cycle and then RMS of that error), and then this RMSE was averaged across all cycles for that participant.  In Table 5, there is no mean+/-SD for the %RMSE since stride length is a single variable, so it is assumed that RMSE in this case is across all trials.  It may be helpful to point out how these calculations were achieved.

·       Line 188: “In overall” should be “Overall”

·       Line 189: “were of” should be “were”

·       Line 192: “overcome” should be “overcomes”

·       Line 196: “used IMU” should be “used an IMU”

·       Line 199: “more” should be “longer”

·       Line 211: Please reword this sentence to indicate that the gait pattern may not be represented by the inverted pendulum model.

·       Line 213: Please provide more information on how personalization could be achieved and at what cost (resources, time, feasibility, etc.).

·       Line 217-218: Consider “… algorithm allow a decrease in the average error and the dispersion by a factor of 2.5 compared to …”

·       Line 224: “to linear” should be “to a linear”

·       Line 240: There should be a period after “unusable [27]”

·       Line 241: “an” should be removed

·       Line 250: “in” should be “on”

·       Line 251: insert a comma after “For attitude estimation”

·       Line 252: “filter suit” should be “filter is suitable for”

·       Line 254-259: This paragraph does not seem necessary.

Author Response

Response to Reviewer 2 Comments

We express our sincere thanks to the reviewer for his/her careful reading and extensive review of the manuscript. The comments were very helpful and constructive. The manuscript has been considerably improved following these remarks.

The responses at each reviewer’s comment are given hereafter. The modifications are highlighted in the manuscript.

1. My one major concern is that only three participants were used as test subjects, and one of them only had four gait cycles.  I think this needs to be addressed more comprehensively in the limitations section.  What are the implications for having such a small test group and limited sample from the individual with the greatest error?

Thank you for this relevant comment. Even if the robustness of the proof is directly related to the number of participants included in the study, this relatively small number must be put in perspective compared to existing published datasets.

Indeed, such a small population is not uncommon in studies focusing on a first assessment of a prosthesis design on people with amputation. The following references are examples of such studies (TT stand for Transtibial amputation and TF for transfemoral amputation) : [1] 5TT, [2] 6TT, [3] 1TF, [4] 2TF, [5] 1TF, [6] 5TF, [7] 1TF, [8] 3TF. .

1. Mariani, B. Assessment of Foot Signature Using Wearable Sensors for Clinical Gait Analysis and Real-Time Activity Recognition. Phd Thesis, École polythechnique fédérale de Lausanne: Lausanne, Switzerland, 2012.

2. Stolyarov, R.; Burnett, G.; Herr, H. Translational Motion Tracking of Leg Joints for Enhanced Prediction of Walking Tasks. IEEE Trans. Biomed. Eng. 2018, 65, 763–769.

3. Seel, T.; Raisch, J.; Schauer, T. IMU-Based Joint Angle Measurement for Gait Analysis. Sensors 2014.

4. Seel, T.; Graurock, D.; Schauer, T. Realtime assessment of foot orientation by Accelerometers and Gyroscopes. Curr. Dir. Biomed. Eng. 2015.

5. Lawson, B.E.; Varol, H.A.; Goldfarb, M. Standing stability enhancement with an intelligent powered transfemoral prosthesis. IEEE Trans. Biomed. Eng. 2011.

6. Huang, H.; Zhang, F.; Hargrove, L.J.; Dou, Z.; Rogers, D.R.; Englehart, K.B. Continuous locomotion-mode identification for prosthetic legs based on neuromuscular - Mechanical fusion. IEEE Trans. Biomed. Eng. 2011, 58, 2867–2875.

7. Varol, H.A.; Sup, F.; Goldfarb, M. Multiclass real-time intent recognition of a powered lower limb prosthesis. IEEE Trans. Biomed. Eng. 2010.

8. Lawson, B.E.; Mitchell, J.; Truex, D.; Shultz, A.; Ledoux, E.; Goldfarb, M. A robotic leg prosthesis: Design, control, and implementation. IEEE Robot. Autom. Mag. 2014, 21, 70–81.

Thus, in our sample, a significant effect of the individual has been observed through one way analysis of variance, showing that the results should be regarded as preliminary results. However, the recruitment of people with amputation is a real issue which limits the possibility of extensive study.

Despite these limitations, this preliminary study remains useful and valuable as it is performed on people with amputation and the signals recorded by the IMU therefore present the characteristics relative to this specific population. In particular, the absence of soft tissues implies direct transmission of mechanical load in the prosthesis (made mainly of carbon fiber and aluminum). During gait the impact are therefore transmitted entirely to the IMU leading to different needs in the filtering process than for signals obtained on non-amputee subjects. This justifies the choice to not include other participants such as non-amputee subjects easier to recruit but that will not lead to realistic signals in the context of prosthetic design.

The first participant had lower walking ability compared to the other two, and decided to stop the acquisitions due to tiredness, hence he only completed 4 gait cycles. These 4 cycles account for 11% of all cycles used in this study, they have thus only a limited effect on the final results.

The limitation part of the article was modified to include these remarks as follows:

The evaluation of the algorithms was based on data obtained from only 3 people with transfemoral amputation. This small number of participants is due to the difficulties to recruits people with transfemoral amputation presenting good walking abilities (no walking aid). Due to tiredness (from other acquisitions) of the 1st participant acquisitions were stopped, hence he only completed 4 gait cycles. The conclusions presented herein are tainted with individual bias and should be taken as preliminary results.

2. Also, in Line 209 it is stated that this is a “good estimation” – please define good and indicate the implications of error.  How does error influence the gait of people with a lower limb prosthesis?  How much error can be tolerated?

Thank you for this relevant comment. At this line the term “good estimation” refers to the fact that since the errors with the a posteriori correction are close to the literature, the model used during the stance phase does not add bias compared to one point estimation of the velocity. It was unclear in the text and this part has been modified as follow:

Compared to the literature, the a posteriori correction gives results similar to ZVU strategies and close to the adjusted results of Yang & Li, which suggests that the model on which the trajectory estimation is based is a good alternative to the ZVU when the IMU is placed on the tibia.

Regarding the general effects of the results on the gait of people with lower limb amputation, it is difficult to extract a criterion of good trajectory estimation since it depends on the purpose of this estimation. The simplest application would be the estimation of the stride length on the prosthesis in order to gives feedback to the user, in that case the estimation accuracy is not critical and 5-10 cm error with a null average could probably be acceptable. The accuracy of the estimation becomes more critical if this estimation is used as part of the control. For instance it can be used to assess the slope of the walking surface in real time, in order to adapt the prosthesis position accordingly. In that case, an error on the vertical trajectory (the more critical) of 25px is around 1% error on the slope estimation. Though the accuracy needed for the slope estimation is dependent on the adaptation method, an estimation bounded to 3 to 4% should allow adaptation. The authors think that such accuracy could be reached using better sensors and/or higher sampling rate.

3. Line 55 and 178: Consider “… variable gain, and 2/ to assess …”·       Line 140: Consider indicating here how many gait cycles for each participant.  At first it is unclear if it is 36 for each or 12 for each (for a total of 36).  But then in the table on the next page it shows a variable number of gait cycles for each.  It would be good to include this information in this line.

Thank you for this comment, this paragraph has been modified to take your remark into account as follows:

These algorithms have been applied on the data of three people with transfemoral amputation following a protocol approved by the Ethics Committee (Comité de Protection des Personnes CPP NX060336). 4, 20 and 12 gait cycles were extracted for participants 1, 2 and 3 respectively. Hence a total of 36 gait cycles have been considered.

4. Line 144: What does “10PG”, “11TG” and “12JG” mean?  Are they subject IDs?  For the purpose of the publication, it would be best to call them participant 1, 2, and 3.  If the numbers/letters stand for something relevant to the paper, those abbreviations should be defined.  Also, in this table they are referred to as “patients” but “participants” in other tables, and “people” at other times in the text.  It would help to be consistent.

These were subjects IDs and as suggested they have been replaced by participant N°1,2 and 3. The word Patient was a remnant of a previous version of the article, it was changed to Participant. In order to be more consistent, the words volunteers and people were also changed to participants at lines 145 and 148 respectively. We kept the terms people with amputation elsewhere in the text because at these times the fact that the participants are transfemoral amputees is of importance.

5. Line 165: It seems a little odd to include 95% confidence intervals for something with just four cycles.  Is it possible to somehow show all four cycles for this participant instead?

We fully agree with the remark of the reviewer that, it is odd to show 95% confidence interval for such a little sample. We therefore have modified the results given in the tables giving min and max error instead of standard deviations. As concerned curves, the choice of drawing corridors is justified for the sake of the clarity of the figure and to keep the visual presentation of the variability of the results across the different participants. To answer to the reviewer relevant remark, we therefore modified the corridor as the envelope of all the curves of each participant rather than average +/- standard deviation.

6. Line 170: In this table, I assume that the RMSE was calculated over the entire cycle (i.e. error determined for all points in the cycle and then RMS of that error), and then this RMSE was averaged across all cycles for that participant.  In Table 5, there is no mean+/-SD for the %RMSE since stride length is a single variable, so it is assumed that RMSE in this case is across all trials.  It may be helpful to point out how these calculations were achieved.

Thank you for this comment. The RMSE on angles and trajectories were indeed computed over each cycle whereas the %RMSE on the stride length was computed across all cycles. Since the %RMSE was used for comparison with only one reference containing other error assessment, this error was removed from the results and the mean error was used instead. The “data analysis” section has been rewrite as follow.

For the orientation, the Root Mean Square Errors (RMSE) of the estimation of the attitude (roll and pitch) provided by the method described in this article relative to the reference has been computed for each gait cycle. Similarly, we computed the RMSE of the trajectory of the knee center. Average, minimum and maximum were then calculated across all gait cycles for each participant.

 In order to compare our results to the literature, we compute the difference between the estimations of the stride length from the knee trajectory, obtained with the IMU and with the reference, at each gait cycle. This error was then normalized using the reference estimation and averaged across all gait cycles.

7. Line 211: Please reword this sentence to indicate that the gait pattern may not be represented by the inverted pendulum model.

Thank you for this comment, the sentence has been modified as follow:

This can be explained by the fact that this participant had a very particular gait pattern that might not be represented by the inverted pendulum rolling on a curved arc foot model especially when using an average round shape of the curved arc foot.

8. Line 213: Please provide more information on how personalization could be achieved and at what cost (resources, time, feasibility, etc.).

Thank you for this comment, this paragraph has been modified to give information regarding personalization of the model as follows:

Personalization of this model would probably result in a more accurate trajectory estimation. For a prosthetic foot this round shape mainly depends on the prosthetic foot design and its alignment [16], personalization could therefore be made using the characteristics of the prosthetic foot and a measurement of the foot alignment.

9.  Line 254-259: This paragraph does not seem necessary.

We agree that this paragraph is a bit redundant but we think it is useful to synthetize the content of the paper.

10. Wording remarks:

Thank you for the wording remarks. All the following has been taken into account.

·       Line 47: “The” should be “the”

·       Line 52: “technics” should be “techniques”

·       Line 147 and 229: the $ should go before the number ($0.80 is common).

·       Line 155: “relatively” should be “relative”

·       Line 165: “Stride” should be “stride”

·       Line 188: “In overall” should be “Overall”

·       Line 189: “were of” should be “were”

·       Line 192: “overcome” should be “overcomes”

·       Line 196: “used IMU” should be “used an IMU”

·       Line 199: “more” should be “longer”

·       Line 217-218: Consider “… algorithm allow a decrease in the average error and the dispersion by a factor of 2.5 compared to …”

·       Line 224: “to linear” should be “to a linear”

·       Line 240: There should be a period after “unusable [27]”

·       Line 241: “an” should be removed

·       Line 250: “in” should be “on”

·       Line 251: insert a comma after “For attitude estimation”

·       Line 252: “filter suit” should be “filter is suitable for”

Reviewer 3 Report

The paper has the potential for further development and can be applied to various contexts associated with human activity processing. However, there are still several problems to be solved in this paper.

1.Some word spelling errors should be fixed.

2.The presented idea is not convincing as the sample size is small in the paper.

3.The experiment results are not clear enough and the comparison is simpler.

4.The related work could be extended and incorporates more comprehensive discussions. The following references maybe helpful to you.

Anderson, David B, Mathieson, Stephanie, Eyles, Jillian, et al. Measurement properties of walking outcome measures for neurogenic claudication: a systematic review and meta analysis. Official journal of the North American Spine Society, SN:1878-1632, 2018.

Liu L, Wang S, Hu B, Wen JH, Qiong QY, Rosenblum, DS. Learning Structures of Interval-Based Bayesian networks in Probabilistic Generative Model for Human Complex Activity Recognition. Pattern Recognition, 81:545-561, 2018. 

Author Response

Response to Reviewer 3 Comments

We express our sincere thanks to the reviewer for his/her careful reading of the manuscript and his/her very helpful and constructive comments. The manuscript has been considerably improved following these remarks. The following regroup the response for each comment made by the reviewer. The modifications are highlighted in the manuscript.

1. Some word spelling errors should be fixed.

Thank you for your comment, some errors have been fixed. Please let us know if any have been missed.

2. The presented idea is not convincing as the sample size is small in the paper.

Thank you for this relevant comment. Even if the robustness of the proof is directly related to the number of participants included in the study, this relatively small number must be put in perspective compared to existing published datasets.

Indeed, such a small population is not uncommon in studies focusing on a first assessment of a prosthesis design on people with amputation. The following references are examples of such studies (TT stand for Transtibial amputation and TF for transfemoral amputation) : [1] 5TT, [2] 6TT, [3] 1TF, [4] 2TF, [5] 1TF, [6] 5TF, [7] 1TF, [8] 3TF.

1. Mariani, B. Assessment of Foot Signature Using Wearable Sensors for Clinical Gait Analysis and Real-Time Activity Recognition. Phd Thesis, École polythechnique fédérale de Lausanne: Lausanne, Switzerland, 2012.

2. Stolyarov, R.; Burnett, G.; Herr, H. Translational Motion Tracking of Leg Joints for Enhanced Prediction of Walking Tasks. IEEE Trans. Biomed. Eng. 2018, 65, 763–769.

3. Seel, T.; Raisch, J.; Schauer, T. IMU-Based Joint Angle Measurement for Gait Analysis. Sensors 2014.

4. Seel, T.; Graurock, D.; Schauer, T. Realtime assessment of foot orientation by Accelerometers and Gyroscopes. Curr. Dir. Biomed. Eng. 2015.

5. Lawson, B.E.; Varol, H.A.; Goldfarb, M. Standing stability enhancement with an intelligent powered transfemoral prosthesis. IEEE Trans. Biomed. Eng. 2011.

6. Huang, H.; Zhang, F.; Hargrove, L.J.; Dou, Z.; Rogers, D.R.; Englehart, K.B. Continuous locomotion-mode identification for prosthetic legs based on neuromuscular - Mechanical fusion. IEEE Trans. Biomed. Eng. 2011, 58, 2867–2875.

7. Varol, H.A.; Sup, F.; Goldfarb, M. Multiclass real-time intent recognition of a powered lower limb prosthesis. IEEE Trans. Biomed. Eng. 2010.

8. Lawson, B.E.; Mitchell, J.; Truex, D.; Shultz, A.; Ledoux, E.; Goldfarb, M. A robotic leg prosthesis: Design, control, and implementation. IEEE Robot. Autom. Mag. 2014, 21, 70–81.

Thus, in our sample, a significant effect of the individual has been observed through one way analysis of variance, showing that the results should be regarded as preliminary results. However, the recruitment of people with amputation is a real issue which limits the possibility of extensive study.

Despite these limitations, this preliminary study remains useful and valuable as it is performed on people with amputation and the signals recorded by the IMU therefore present the characteristics relative to this specific population. In particular, the absence of soft tissues implies direct transmission of mechanical load in the prosthesis (made mainly of carbon fiber and aluminum). During gait the impact are therefore transmitted entirely to the IMU leading to different needs in the filtering process than for signals obtained on non-amputee subjects. This justifies the choice to not include other participants such as non-amputee subjects easier to recruit but that will not lead to realistic signals in the context of prosthetic design.

The limitation part of the article was modified to include these remarks as follows:

The evaluation of the algorithms was based on data obtained from only 3 people with transfemoral amputation. This small number of participants is due to the difficulties to recruits people with transfemoral amputation presenting good walking abilities (no walking aid). Due to tiredness (from other acquisitions) of the 1st participant, acquisitions were stopped, hence he only completed 4 gait cycles. The conclusions presented herein are tainted with individual bias and should be taken as preliminary results.

3. The experiment results are not clear enough and the comparison is simpler.

Thank you for this comment, in order for the results to be clearer, the part of the materials and methods related to the definition of the computed errors was reworked as follows. Please note that the %RMSE on the stride length estimation was removed in order to suppress confusion between RMSE on the full trajectory computed for each cycle and the RMSE on the stride length computed across cycles.

For the orientation, the Root Mean Square Errors (RMSE) of the estimation of the attitude (roll and pitch) provided by the method described in this article relative to the reference has been computed for each gait cycle. Similarly, we computed the RMSE of the trajectory of the knee center. Average, minimum and maximum were then calculated across all gait cycles for each participant.

 In order to compare our results to the literature, we compute the difference between the estimations of the stride length from the knee trajectory, obtained with the IMU and with the reference, at each gait cycle. This error was then normalized using the reference estimation and averaged across all gait cycles.

Regarding comparison to the literature, studies developing algorithms for the estimation of orientation from IMUs usually evaluate their algorithm only through RMSE as in the following references.

Valenti, R.G.; Dryanovski, I.; Xiao, J. Keeping a good attitude: A quaternion-based orientation filter for IMUs and MARGs. Sensors (Switzerland) 2015.

Madgwick, S.O.H.; Harrison, A.J.L.; Vaidyanathan, R. Estimation of IMU and MARG orientation using a gradient descent algorithm. In Proceedings of the 2011 IEEE International Conference on Rehabilitation Robotics; IEEE: Zurich, 2011; pp. 1–7.

Tian, Y.; Wei, H.; Tan, J. An adaptive-gain complementary filter for real-time human motion tracking with MARG sensors in free-living environments. IEEE Trans. Neural Syst. Rehabil. Eng. 2013.

For trajectory the authors is not aware of any study reporting error on the full trajectory, we thus choose to compute the RMSE to stay consistent with the orientation errors computation.

In the literature most of the studies evaluating algorithms for strapdown integration use an error on the stride length estimation or on the total distance walked as in the references below, we thus compute similar errors.

Mariani, B.; Hoskovec, C.; Rochat, S.; Büla, C.; Penders, J.; Aminian, K. 3D gait assessment in young and elderly subjects using foot-worn inertial sensors. J. Biomech. 2010, 43, 2999–3006

Yun, X.; Bachmann, E.; Moore, H.; Calusdian, J. Self-Contained Position Tracking of Human Movement Using Small Inertial/Magnetic Sensor Modules. In Proceedings of the 2007 IEEE International Conference on Robotics and Automation; Roma, Italy, 2007; pp. 2526–2533.

Yang, S.; Li, Q. IMU-based ambulatory walking speed estimation in constrained treadmill and overground walking. Comput. Methods Biomech. Biomed. Engin. 2012.

4. The related work could be extended and incorporates more comprehensive discussions. The following references maybe helpful to you.

Anderson, David B, Mathieson, Stephanie, Eyles, Jillian, et al. Measurement properties of walking outcome measures for neurogenic claudication: a systematic review and meta analysis. Official journal of the North American Spine Society, SN:1878-1632, 2018.

Liu L, Wang S, Hu B, Wen JH, Qiong QY, Rosenblum, DS. Learning Structures of Interval-Based Bayesian networks in Probabilistic Generative Model for Human Complex Activity Recognition. Pattern Recognition, 81:545-561, 2018.

Thank you for this remark and for the interesting references. Results were indeed given with a specific application in mind, whereas this algorithms could be used in other fields. The paragraph below was added at the end of the discussion in order to address this. Only the first given reference were used as the second is much more high level (in the programming sense), thus the links with the presented study seems a bit distant.

The presented algorithm is primarily designed for prostheses control (i.e. real time activity recognition), but it could also be useful for orthosis, exoskeleton, or activity monitoring device. This algorithm could also be adapted for an IMU placed more proximally with a modification of the model of the kinematics of the unipodal stance phase. Moreover its application range is not limited to activity recognition, it could also be used for activity monitoring of specific pathologies as suggested in [34].

Round 2

Reviewer 1 Report

The authors carefully revised the manuscript and addressed all issues raised with respect to the previous version in a satisfactory way. The updated manuscript is now satisfactory in all aspects.

Author Response

We deeply thank the reviewer for his/her examination of the manuscript and kind comments on the presented manuscript. For a better understanding some modification of the results part have been made. These modifications are highlighted in revision mode in the new version of the manuscript.

Reviewer 3 Report

    This paper presents new ideas on gait pattern recognition and provides an adequate introduction to background and relevant work. However, there are still some problems to be solved before publication.

In the two chapters of the results and discussion, the format is confusing, such as lines 169 and 236.

In the results section, estimates of reference and their own method are provided, but it does not indicate what method of reference was used, and there is no explanation of the contents of the chart (like Figure 5, the gait cycle is all or a single), and it is not standard and difficult to understand. 

Author Response

Thank you very much for your review comments. Please find our response at the attachment.
